Pattern classification of brain activation during emotional processing in subclinical depression: psychosis proneness as potential confounding factor

Modinos Gemma gemma.modinos@kcl.ac.uk 1
Mechelli Andrea 1
Pettersson-Yeo William 1
Allen Paul 1
McGuire Philip 1
Aleman Andre 2
1 Department of Psychosis Studies, Institute of Psychiatry, King’s College London , London , United Kingdom
2 Department of Neuroscience, Neuroimaging Center (NIC), University Medical Center Groningen, University of Groningen , The Netherlands
Iacoboni Marco
Electronic publication date: 2013 Feb 26
Publication date: 2013
Volume: 1
Electronic Location ID: e42
Received 2012 Dec 4; Accepted 2013 Jan 28
Copyright: © 2013 Modinos et al.
Copyright year: 2013
Copyright holder: Modinos et al.
License: This is an open access article distributed under the terms of the Creative Commons Attribution License, which permits unrestricted use, distribution, and reproduction in any medium, provided the original author and source are credited.
License URL: https://creativecommons.org/licenses/by/3.0/

Keywords: Machine learning, Support vector machine, fMRI, Emotion, Subclinical depression, Psychosis proneness, Neuroimaging

Funding: NARSAD Independent Investigator Award European Science Foundation EURYI grant NWO number: 044035001 Andrea Mechelli is supported by a NARSAD Independent Investigator Award. Data collection was supported by a European Science Foundation EURYI grant (NWO number: 044035001) awarded to André Aleman. The funders had no role in study design, data collection and analysis, decision to publish, or preparation of the manuscript.

==============================
We used Support Vector Machine (SVM) to perform multivariate pattern classification based on brain activation during emotional processing in healthy participants with subclinical depressive symptoms. Six-hundred undergraduate students completed the Beck Depression Inventory II (BDI-II). Two groups were subsequently formed: (i) subclinical (mild) mood disturbance (n = 17) and (ii) no mood disturbance (n = 17). Participants also completed a self-report questionnaire on subclinical psychotic symptoms, the Community Assessment of Psychic Experiences Questionnaire (CAPE) positive subscale. The functional magnetic resonance imaging (fMRI) paradigm entailed passive viewing of negative emotional and neutral scenes. The pattern of brain activity during emotional processing allowed correct group classification with an overall accuracy of 77% (p = 0.002), within a network of regions including the amygdala, insula, anterior cingulate cortex and medial prefrontal cortex. However, further analysis suggested that the classification accuracy could also be explained by subclinical psychotic symptom scores (correlation with SVM weights r = 0.459, p = 0.006). Psychosis proneness may thus be a confounding factor for neuroimaging studies in subclinical depression.

Introduction

Abnormalities in the brain circuitry underlying emotional processing may be key in determining vulnerability to major depressive disorder (MDD) (Davidson et al., 2002). Neuroimaging studies of healthy volunteers have identified a neural circuitry important for the experience of affective states, involving the amygdala, insula, anterior cingulate cortex (ACC), orbitofrontal and medial prefrontal cortex (PFC) (Phillips et al., 2003a). In MDD, studies have documented abnormal activity within this network compared to healthy controls during emotional experience (Phillips et al., 2003b). It is unclear to what extent these abnormalities can be considered a marker of the disorder and to what extent they can be considered a marker of vulnerability. This can be investigated by examining brain activation during emotional experience in young adults with subclinical depressive symptoms.

Studies in groups with subclinical symptomatology should consider that alterations in brain activation in subclinical groups are likely to be subtler than those observed in full-blown disorders. The standard approach in the analysis of functional magnetic resonance imaging (fMRI) data is based on the General Linear Model (GLM) (Friston et al., 1995), and is known as mass-univariate because it makes statistical inferences in each location (voxel) independently. However, fMRI data are multivariate in nature since each scan contains information about brain activation at thousands of measured locations (voxels). Based on multivariate statistics, a multivariate analysis may thus be more sensitive to spatially distributed and subtle effects in the brain than a standard mass-univariate analysis, potentially providing a more powerful approach for studies of subclinical populations in which less severe alterations are generally observed. An additional advantage is that it allows inferences to be made at the level of the individual rather than the group and therefore has high translational potential in a clinical setting. The Support Vector Machine (SVM) (Vapnik, 1995) is a powerful tool for statistical pattern classification through which the combination of all voxels as a whole is identified as a global spatial pattern by which the groups differ. In MDD, the application of multivariate analysis to MRI scans has yielded promising results, achieving diagnostic classification accuracies of 67%–86% with functional and 68%–85% with structural MRI (reviewed in Orrù et al., 2012). The application of SVM in MDD has a clear translational potential by identifying biomarkers that allow prediction of treatment response at the individual level. In addition, machine learning weighting factors (used for making predictions) based on abnormalities of brain structure can inform clinical practice reflecting an objective biomarker of MDD illness severity, as recently shown by Mwangi et al. (2012) who achieved 90% classification accuracy with structural MRI scans of MDD patients and controls and observed the SVM weighting factors correlated strongly with subjective ratings of illness severity.

Interestingly, Fu et al. (2008) used an emotional processing fMRI task involving sad facial stimuli in a sample of 19 patients with MDD and 19 healthy controls and reported successful discrimination between the groups with an overall accuracy of 86% (p < 0.001). However, no previous studies have applied multivariate techniques to address the issue of whether these neurobiological characteristics relate to disease vulnerability. Our group has previously reported significant SVM classification (69.4% accuracy, p = 0.017) in a subclinical population, that is, in individuals with psychosis proneness, using functional activation during emotional processing (Modinos et al., 2012). No studies to date have applied multivariate analysis to a sample of individuals with subclinical depression. We therefore used SVM to examine whether individuals with subclinical depressive symptoms would show different brain activation during emotional processing using a multivariate approach. We sought to determine (i) whether a feature classifier based on functional parameters during emotional processing could reliably discriminate between participants with, and without, subclinical depressive symptoms, (ii) which regions contributed to this discrimination and (iii) the role of confounding factors, in particular of subclinical psychosis proneness, as this has been shown to be associated depression (Verdoux et al., 1999) and with emotional processing abnormalities (van ’t Wout et al., 2004).

Materials and methods

Participants

Six hundred undergraduate students completed the Beck Depression Inventory II (BDI-II) (Beck et al., 1996). Of these, 17 (10 female) scored ∼14 (range 11-19, mild mood disturbance) and agreed to take part in the study. An age- and gender-matched control group of 17 students (10 female) with low scores (no mood disturbance) was recruited to ensure matching of demographic variables (see Table 1).

Table 1 Characteristics of study participants.

	Subclinical depression group	Control group	P value	
Age (years)	20.5 (Range: 18-27; SD 2.4)	20.7 (Range: 18-27; SD 2.3)	0.886	
Gender (percentage Female)	10 (%58.8)	10 (%58.8)		
Beck Depression Inventory-II (BDI-II)	14.2 (Range: 11-19; SD 2.9)	2.1 (Range: 0-9; SD 2.4)	 < 0.001	
Community Assessment of Psychic Experiences (CAPE)	1.5 (Range: 1.2-1.8; SD 0.2)	1.1 (Range: 1-1.2; SD 0.04)	 < 0.001	

The BDI-II is a 21-item mood scale designed to measure depressive symptoms, with each item scored on a four-point scale. No participants scoring above BDI-II Major Depressive Disorder cut-off ( ≥ 20) were included. Participants also completed the Community Assessment of Psychic Experiences (CAPE) (Stefanis et al., 2002), a self-report instrument on psychotic symptoms administered as part of a parallel study on emotional processing in psychosis proneness. In the present study, we used the CAPE to examine the effect of psychosis proneness as a possible confounding variable. Subjects were screened for exclusion criteria by experienced raters using a self-report checklist for healthy participants, comprising (i) no personal history of neurological or psychiatric illness, (ii) no family history of psychotic or neurological illness in first-degree relatives, (iii) no use of illicit substances, and (iv) no changes in overall level of functioning over the past 6 months. After subjects were given a complete explanation of the study, written informed consent was obtained from all of them. Participants were paid for their participation. The study was approved by the Medical Ethical Committee of the University Medical Center Groningen, and was conducted in accordance with the Declaration of Helsinki.

Emotional processing task

The stimulus set consisted of 66 colour pictures from the International Affective Picture System (IAPS) (Lang, Bradley & Cuthbert, 1997). Twenty-two neutral (mean valence ±  SD = 5.1 ± 1.7; mean arousal ± SD = 2.89 ± 2) and 44 negative pictures (mean valence ± SD = 2.5 ± 1.6; mean arousal ±  SD = 5.8 ± 2.2) were chosen based on normative ratings, according to which 9 represents a high rating on each dimension (e.g., high arousal, positive valence), and 1 represents a low rating on each dimension (e.g., low arousal, negative valence) (Lang, Bradley & Cuthbert, 1997; Lang, Bradley & Cuthbert, 2008). All negative pictures depicted complex scenes of burn victims, funerals, and interpersonal violence, and were matched for arousal ratings and visual complexity. The task design has been described in detail elsewhere (Modinos, Ormel & Aleman, 2010). In brief, at the beginning of each trial, a photo was presented in the centre of a black screen, for 2 s, with the instruction VIEW displayed in white letters underneath, for emotional induction. During this period subjects were to view the photo and allow themselves to naturally experience any emotional response to it. The fMRI paradigm involved 2 main conditions: Negative (viewing of a negative picture), and Neutral (viewing of a neutral picture, serving as control condition). For the purpose of this study, we focused on the initial 2 s of emotional induction. The rest of a trial-block involved: 4 s of either attending or cognitively down-regulating their emotional response to the photo, 3.1 s of a fixation cross, 3 s for rating the strength of their negative emotion with a button response, and 5 s with the word “RELAX” before the next trial began, with a constant gap between trials, as explained in Modinos, Ormel & Aleman (2010). In its entirety, the experimental paradigm comprised 66 trials of 18 s, interleaved with four 20-s rest trials (fixation cross). All participants underwent a training session immediately before fMRI scanning with a different set of 10 IAPS pictures.

fMRI measurements and preprocessing

Images were acquired on a 3-T Philips Intera MR scanner (Philips Medical Systems, Best, the Netherlands). Functional MRI data comprised 634 volumes acquired with T2*-weighted gradient echo-planar imaging (EPI) sequences, using a sense-8 head coil, in two functional runs of 317 volumes. Thirty-seven slices per volume, sensitive to blood oxygenation level-dependent (BOLD) contrast, were obtained using a TR of 2 s, flip angle = 70∘, TE = 35 ms; inplane resolution = 3.5 × 3.5 mm, and field of view = 224 mm. Slices were acquired interleaved and oriented parallel to the AC–PC plane, with a thickness of 3.5 mm and no gap. High-resolution T1-weighted 3D fast-field echo sequences were obtained for anatomical reference (160 slices, TR = 25 ms, TE = 4.6 ms, slice thickness = 1 mm; matrix size = 256 × 256; field of view = 260 mm; voxel size = 1 × 1 × 1 mm).

Image preprocessing was carried out in SPM5 (www.fil.ion.ucl.ac.uk/spm). All functional images were slice-time corrected, and realigned. After realignment, the obtained mean EPI image was co-registered with the structural T1 image. Subsequently, images were spatially normalised to the standard stereotactic space defined by the Montreal Neurological Institute (MNI) template. During normalisation, scans were re-sampled onto a 2 × 2 × 2 mm3 grid. Functional images were spatially smoothed with a 3D isotropic Gaussian kernel (FWHM of 8 mm). Low-frequency noise was removed by applying a high-pass filter (cut-off of 128 s) to the fMRI time-series at each voxel. Statistical parametric maps during emotional experience in each subject were identified with the Negative > Neutral contrast. Effects were modelled using a boxcar convolved with a canonical hemodynamic response function for the 2 s trial epoch during which participants viewed each picture.

Pattern classification analysis

SVM was implemented on the Pattern Recognition of Brain Image Data toolbox (Probid; http://www.brainmap.co.uk/probid.htm). Multivariate analyses were carried out as detailed in our previous study (Modinos et al., 2012). Individual Negative > Neutral contrast images were treated as points located in a high dimensional space defined by the voxel values. A linear decision boundary in this high dimensional space was defined by a “hyperplane” that separated the individual contrast images according to a class label (i.e. subclinical depression, controls). The optimal separating hyperplane was computed based on the multivariate pattern of voxel values across each contrast image. This hyperplane can be described in terms of a vector of voxel weights, plus an offset. In order to project each subject’s map onto the weight vector, and thus observe their relative distance from the hyperplane, the inner product is obtained between the weight vector and that subject’s input vector, and the result added to the offset. This provides a single absolute decision value for each subject indicating their relative distance from the hyperplane, and thus the relative ease, or difficulty, with which they were classified. While the absolute value of weight vector scores depends on several methodological variables and is not meaningful per se (Lee et al., 2010), it can be used to identify voxels which provide the greatest relative contribution to classification (Orrù et al., 2012). The minimal distance from the separating hyperplane to the closest training example is called the margin. The training examples that lie on the margin are called support vectors. They are conceptually the most difficult data points to classify and therefore they define the location of the separating hyperplane. The optimal hyperplane has been shown to be the one with maximal margin (i.e., more separation between the classes), so that a larger margin reduces risk of group overlap corresponding to a better generalisation. The SVM thus selects from many possible solutions the optimal one, which is determined by the most difficult to classify, hence most informative, training examples (the support vectors). Advantages of this method are the selection of training examples that are most informative for the classification and a good scaling for high dimensions. The SVM approach considers all brain voxels jointly in a high-dimensional space rather than examining each brain region individually (Lao et al., 2004). As the classifier is multivariate by nature, the combination of all voxels as a whole is identified as a global spatial pattern by which the groups differ (i.e. the discriminating pattern). The PROBID software allows a linear kernel matrix (measuring similarity between all pairs of brain images) to be pre-computed and supplied to the classifier. A linear rather than a non-linear kernel SVM was used in order to reduce the risk of overfitting the data and to allow direct extraction of the weight vector as an image (i.e. the SVM discrimination map). This approach affords a substantial increase in computational efficiency and permits whole-brain classification without requiring explicit dimensionality reduction. A “leave-one-out” cross-validation method was used (Lemm et al., 2011), which involved excluding a single subject from each group and training the classifier using the remaining subjects; the subject pair excluded was then used to test the ability of the classifier to reliably distinguish between categories. SVM classification in PROBID is provided by the LIBSVM library (http://www.csie.ntu.edu.tw/ cjlin/libsvm/). At the end of the process, a multivariate discrimination map was generated visualising the most discriminating regions between groups.

Based on our a priori hypothesis of differential response to negative pictures within a network of regions involved in emotional processing, we created an anatomical mask which included the bilateral amygdala, insula, anterior cingulate cortex (ACC), orbitofrontal, and medial PFC using the Automated Anatomical Labeling (AAL) as implemented in the WFU_Pickatlas toolbox. These regions were chosen based on the literature on affective neuroscience in the healthy brain (Davidson & Irwin, 1999; LeDoux, 1996; Phan et al., 2002; Phillips et al., 2003a), and evidence of abnormalities in patients with psychosis and at-risk populations (Aleman & Kahn, 2005; Brunet-Gouet & Decety, 2006; Phillips & Seidman, 2008; Phillips et al., 2003b). We then used this mask to constrain the search of significant group differences for the comparison between negative and neutral pictures. In addition to this region-of-interest (ROI) analysis, we conducted an SVM analysis of the whole brain minus our regions of interest for completeness.

In order to examine whether the differential group activations observed with SVM would also be seen with a standard GLM analysis, we applied a paired t-test design in SPM5 using the same contrast images (Negative > Neutral) and the same subject pairs used for the SVM classification. The same pre-defined anatomical “emotional” mask was used to constrain the search of significant group differences for the comparison between negative and neutral pictures. Finally, as this was an implicit emotion processing paradigm and no behaviour was recorded from the subjects inside the scanner, we conducted a within-group GLM analysis to examine whether the task induced the expected effects within each group. Statistical inferences were made at p < 0.05 after False-Discovery Rate correction (FDR).

Results and discussion

Behavioural results

There were no significant differences in age between the groups (F1,32 < 1; p = 0.886). Participants with subclinical depressive symptoms also showed higher psychotic symptom scores (F1,32 = 82.079; p < 0.001) than controls.

SVM results

The classification sensitivity was 71% (subclinical depression correctly classified), and the specificity 82% (controls correctly classified), corresponding to an accuracy of 77% (see Fig. 1). The set of regions which showed different values between the subclinical depression group and the control group are reported in Table 2. The threshold of classification strength was set to show the top 5% of the weight vector scores, as done in our previous study (Modinos et al., 2012).

Figure 1 Weight vector map showing the most discriminating brain regions between groups (top 5%).

Regions that contributed more to classifying individuals with subclinical depression (n = 17) are shown in red, while regions that contributed more to the classification of controls are shown in blue (n = 17), in axial (A) and coronal (B) views (z = [−22, −17, −12, −2, 3, 18, 28, 38, 48]; y = [−1, 2]). (C) Projection of each subject onto the weight vector, with positive patterns (red circles) discriminating for subclinical depression, and negative patterns (blue crosses) for controls (p = 0.002).

Table 2 Most important activation regions (top 5%) discriminating between each group (subclinical depression and controls).

Anatomical region	Side	MNI coordinates	w i	
		x	y	z		
Controls  > Subclinical depression						
Medial frontal gyrus	L	−3	57	4	13.27	
Medial frontal gyrus	R	4	52	8	11.74	
Anterior cingulate cortex		0	32	21	22.49	
Insula	L	−33	12	−16	35.04	
Insula	R	34	28	3	31.84	
Amygdala	R	30	0	−25	25.12	
Amygdala	L	−24	0	−25	20.60	
Subclinical depression  >  controls						
Medial frontal gyrus	R	8	59	27	28.61	
Medial frontal gyrus	L	−20	39	26	24.31	
Anterior cingulate cortex	R	−4	36	4	8.76	
Insula	R	44	23	3	29.64	
Insula	L	−38	21	2	20.68	
Amygdala	R	28	−3	−15	5.33	
Amygdala	L	−30	−4	−16	4.62	
Notes.

MNI: Montreal Neurological Institute space; L: Left; R: Right; wi: weight of each cluster centroid i.

We performed a permutation test to examine the statistical significance of the classification result. Specifically, we carried out 1000 random assignments of all participants to the “subclinical depression” or “control” categories, and then performed the leave-1-out procedure. By calculating the number of occasions where the permuted accuracy was greater than the obtained accuracy, and dividing this figure by 1000, we were able to estimate the probability that the observed result was obtained by chance. This showed that the statistical significance of the classification obtained in relation to chance was p = 0.002.

The SVM analysis including the whole brain minus our regions of interest did not yield a statistically significant classification (overall accuracy was 56%, p = 0.306; 1000 permutations).

Effect of psychosis proneness

In order to examine the possible contribution of psychosis proneness to the SVM classification, we subjected the individual CAPE scores to a correlation analysis (Pearson’s r) with the projection of each subject’s brain image onto the weight vector (i.e. distance of each test subject’s scan from the classification hyperplane thus quantifying the relative ease, or difficulty, with which they were categorised). This revealed a significant association (r = 0.459, p = 0.006), suggesting that psychosis proneness scores were associated with the neural activity underlying the classification results.

GLM results

Standard univariate between-group analysis revealed no significant differences for the contrast Negative > Neutral either with ROI analysis or with the whole-brain mask minus our regions of interest, corrected for multiple comparisons (p < 0.05). At a more lenient level (p < 0.001, uncorrected), subjects with subclinical depression, compared with controls, showed decreased activation in the left insula (x = −38 z = 8 y = −12, k = 72 voxels), anterior cingulate cortex (x = 6 z = 18 y = 22, k = 39 voxels), and bilaterally in the amygdala (right: x = 20 z = 0 y = −10, k = 7 voxels; left: x = −24 z = −4 y = −12, k = 12 voxels).

Finally, the ROI analysis of the Negative > Neutral contrast within each group showed that the task reliably induced increased activity in these emotion-related regions (namely, medial frontal gyrus including the anterior cingulate cortex, the amygdala and the insula bilaterally), after FDR correction for multiple comparisons (p < 0.05) (Fig. 2, Table 3).

Figure 2 Brain regions showing increased functional activation during the experience of negative emotion (relative to the neutral condition) within groups, as measured with standard General Linear Model analysis and at p < 0.05, FDR corrected (z = −22, −17, −12, −2, 3, 18, 28, 38).

Table 3 Region of interest within-group results with standard GLM analysis on the contrast Negative > Neutral within groups, FDR-corrected for multiple comparisons (p < 0.05).

Anatomical region	Side	MNI coordinates	Number of voxels	t-value	z-value	
		x	y	z				
Controls								
Medial frontal gyrus	L	−8	56	24	2999	10.03	5.57	
Insula	L	−34	16	−16	521	8.73	5.22	
Amygdala	L	−18	−6	−16	204	8.13	5.05	
Amygdala	R	22	0	−20	184	6.82	4.61	
Insula	R	36	16	−16	410	5.77	4.18	
Subclinical depression								
Medial frontal gyrus	L	−8	26	36	512	6.54	4.50	
Medial frontal gyrus	R	6	46	36	288	5.15	3.90	
Insula	R	44	24	−2	27	5.08	3.87	
Insula	L	−32	20	−12	174	5.02	3.83	
Amygdala	L	−20	−8	−16	9	4.07	3.32	
Amygdala	R	22	−6	−14	2	3.23	2.79	
Notes.

MNI: Montreal Neurological Institute stereotactic space, L: Left; R: Right.

Conclusions

The present study suggests that a multivariate approach appears to enable accurate identification of individuals with subclinical depressive symptoms based on brain activation in a network of regions thought to underlie the experience of affective states, comprising the amygdala, insula, ACC, and medial prefrontal cortex bilaterally. Importantly, however, this identification could also be explained by subclinical psychotic symptoms which were associated with depression proneness. In contrast, the standard GLM analysis was unable to detect significant differences in activation between groups, although within-group analyses confirmed that the task successfully induced significant increases in activation in emotional brain regions during the processing of negative emotion in each group.

Our results are in keeping with previous reports of accurate classification of patients with MDD using multivariate analysis methods on functional activation during emotional processing (Fu et al., 2008), and indicate that these methods can also be used to classify individuals with vulnerability to depression. However, associated psychotic symptomatology may have a significant effect on the classification, even at subclinical levels. Indeed, associations between symptoms of depression and psychotic-like experiences in healthy samples have been previously documented (e.g. van ’t Wout et al., 2004; Lewandowski et al., 2006), and may thus warrant being incorporated in the statistical evaluation of brain data. Of note, unlike the present study, in our earlier neuroimaging studies in which the groups were identified based on psychosis proneness scores the results remained significant after controlling for the effects of depressive symptoms (e.g. Modinos, Ormel & Aleman, 2010; Modinos et al., 2012). Although we note that while successful classification of depression proneness could be explained by subthreshold psychotic symptoms in the present investigation, successful classification of psychosis proneness did not appear to be dependent on subthreshold depressive symptoms in a previous study (Modinos et al., 2012). Taken collectively, this pattern of results provides evidence that psychosis proneness was a greater source of influence on brain activation than depression proneness. The application of machine learning approaches to neuroimaging data in psychiatric disorders is relatively young and new developments are in progress to deal with challenges associated with the nature of the populations under study. The present study sought to examine whether associated subclinical psychosis proneness would impact on the classification of MRI images of individuals with subclinical depressive symptoms, and not to control for them. Nevertheless, new methods are being developed to control for confounding variables in SVM analysis of brain data. In Alzheimer’s Disease research, for instance, Dukart, Schroeter & Mueller (2011) proposed a method to control for possible effects of confounding variables such as age prior to statistical evaluation of MRI data using SVM. These type of new advances applied to clinical neurosciences can improve and extend the application of pattern classification algorithms as a means to deal with issues such as concomitant symptomatology.

As a limitation to the study, although all participants denied having had a formal diagnosis of depression, testing our classification algorithm on a clinically diagnosed population while taking into account potential concomitant psychotic symptoms and their effect on the classification should help expand the present findings. Our sample size was sufficient to discriminate between individuals with, and without, subclinical depression scores, with statistically significant accuracy, but it was insufficiently powered to establish, for example, gender differences within groups, which might be important in emotional paradigms (Wager et al., 2003). Finally, while we did not have access to follow-up data in this study, following these subjects over time to elucidate potential transitions to full-blown depression/psychosis would have been of interest. Future studies with a prospective design should provide valuable insights to the use of pattern classification in detecting biomarkers of illness development.

To our knowledge, this is the first study to use a multivariate approach to investigate patterns of brain activation in people with subclinical depressive symptoms. The main finding is that, although a multivariate approach could accurately discriminate these individuals based on a neural network involved in emotional processing, psychosis proneness played a critical role in the classification and may thus be an important confounding factor to be considered in future neuroimaging work in this group.

We thank Anita Kuiper for assistance with MRI scanning, and our study volunteers for their participation.

Additional Information and Declarations

Competing Interests

Author Contributions

Human Ethics

Andrea Mechelli is an Academic Editor for PeerJ.

Gemma Modinos conceived and designed the experiments, performed the experiments, analyzed the data, wrote the paper, participated in interpretation of the data; approved the final version of the manuscript.

Andrea Mechelli and Andre Aleman conceived and designed the experiments, participated in interpretation of the data; approved the final version of the manuscript.

William Pettersson-Yeo analyzed the data, participated in interpretation of the data; approved the final version of the manuscript.

Paul Allen and Philip McGuire participated in interpretation of the data; approved the final version of the manuscript.

The following information was supplied relating to ethical approvals (i.e. approving body and any reference numbers):

The study was approved by the Medical Ethical Committee of the University Medical Center Groningen (Ethical Review Board approval number: METc 2007/105, approved on September 10, 2007), and was conducted in accordance with the Declaration of Helsinki.

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
