# Peer review of "Pattern classification of brain activation during emotional processing in subclinical depression: psychosis proneness as potential confounding factor"

_PeerJ, doi:10.7717/peerj.42_

## Round 0.1 · original submission · Major Revisions

The reviewers have provided constructive criticisms. I do invite you to carefully consider them if you plan to submit a revision of your manuscript.

·

Basic reporting

No comments

Experimental design

I don't have any major comment, only small clarifications:
p.4: If any criterion was used to select the 17 healthy volunteers out of the group of all subjects that scored in the no depression range (e.g., to ensure matching on demographic variables?) it would be worth clarifying.
p.8: As I recall, the importance vector from PROBID gives values for each voxel for group 1 and group 2, so the authors in creating Table 2 must have chosen a threshold of classification strength to define their peaks -- what was that threshold?

Validity of the findings

I only have 2 small issues in this respect:
1. When the authors say "critically this discrimination was based on a network of regions thought to underlie the experience of affective states, comprising the amygdala, insula, ACC and medial prefrontal cortex." Isn't this obvious from the fact that these are all the regions that were chosen in the mask within which classification was performed? My point is that it is important to know not only that regions of this network DO classify, but also that regions extraneous to the network do NOT classify correctly. Perhaps it might be easy for the authors to perform classification on a mask of "1-the network they chose" (i.e., all the regions other than the network).
2. Considering that, as the authors state, the classification result was driven by the psychosis proneness score, isn't it misleading to state that they could successfully classify MMD v healthy volunteers? After all what they did classify, in fact, is psychosis proneness.

Additional comments

Considering the methodological slant the authors give to the paper (e.g., "This is the first time that multivariate analysis are used on MMD ...") it might be interesting (and not too work intensive) to compare the multivariate results to the univariate regression results. It would certainly make the case for using multivariate analysis concrete if indeed it were the case that the differences they find would not be seen with a usual GLM.
As a very small issue, on p.2, perhaps the authors could spell out the first occurrence of "Psychosis Proneness"

·

Basic reporting

The basic reporting seems fine. The manuscript is well-written and clear.

Experimental design

The experimental design seems reasonable in general, but there are places where important details are lacking. For example, the description of the timing of the task and the presentation of the trials is not sufficient to allow replication. Specifically, it would be important to know how long each trial lasted, how much time there was between each trial, whether that gap was constant or jittered, etc.

There are also some important details missing about the statistical analysis. It is not described how the authors assess significance of the classifier results. The results section does mention permutations parenthetically, but a permutation procedure is never described and there is no discussion of statistical thresholding.

The end of the methods section describes the procedure for examining the most discriminating regions between groups. I think this section requires more description. I understand what the SVM weights are, but I do not really understand what it means to "project” onto each subject’s map as described in the results section.

There is a description of the creation of a rather wide-ranging anatomical mask in the methods section but it is never said what is being done with this mask. Are the authors only examining voxel weights within that mask? Is that for statistical reasons? Again, since statistical thresholding is not discussed it is hard for the reader to assess this.

I think it is a weakness of the experimental design that no behavior was recorded from the subjects inside the scanner. We don’t know if the two groups were paying equal attention to the pictures, for example. This increases the number of available explanations for the differences in activation maps.

Validity of the findings

I think the success of the classifier in discriminating between the subclinical depression and the healthy groups is valid and interesting.

I am a little more skeptical however about the correlation with the SVM voxel weights. The meaning of SVM voxels weights is somewhat controversial (see Lee et al., 2010, NeuroImage, for a discussion of some of the issues). My understanding is that weight vectors may have information about localization, but that comparing the magnitudes across studies (or people) may be problematic because they may can influenced by many factors (such as overall signal level, parameters of the SVM, etc.). I’m not a mathematician so this issue is a bit beyond me but I would appreciate at least a further discussion of it in the manuscript. These concerns also might be obviated by a better description of how that analysis was done.

Perhaps more importantly, if the psychotic symptoms score is correlated with the depression score, would it not necessarily be the case that psychotic symptoms are related to classification? I’m not sure what the SVM correlation shows beyond the fact that depression and psychotic symptoms are confounded, which is already revealed by the behavior. (Aren't those with the highest psychosis scores also the most depressed?) A more useful analysis would somehow compare the extent to which these variables contributed to classification, relative to each other.

Relatedly, I do not think the statement at the end of the first paragraph of the conclusion is warranted: “Importantly, however, the results changed after incorporating concomitant subclinical psychotic symptoms in the analysis, suggesting the high classification accuracy obtained was driven by subclinical psychotic symptoms.” This appears to describe an analysis different from the one that was presented in the manuscript, which merely showed an association between psychotic symptoms and classification weights. I don’t see a basis for concluding that psychotic symptoms play a role any more than any number of other variables that might be correlated with depression.

Additional comments

I’m not of the view that multivariate analysis must always be accompanied by univariate analysis, but I think in this case it might be informative. Does the MVPA analysis capture something beyond the group differences in signal magnitude at each voxel? It would be interesting to compare the SVM maps to a univariate group comparison map.

---

## Round 0.2 · accepted · Accept

I think you should consider addressing the remaining comments from the two reviewers. They strike me as valuable additions to your manuscript, and since your manuscript will be published, it is best to publish in its best possible form.

·

Basic reporting

No comments.

Experimental design

No comments

Validity of the findings

No comments

Additional comments

I believe this revision has adequately addressed my concerns, and this version of the manuscript is considerably improved.

The lack of any significant result in the GLM is interesting, although it could just be an issue of power. (Are there any sub-threshold clues in the GLM as to what is driving the difference between the groups in the SVM?)

I apologize to the authors for the incorrect reference; I did indeed intend to refer to the Human Brain Mapping article.

·

Basic reporting

No comments

Experimental design

No comments

Validity of the findings

No comments

Additional comments

Overall I think the new manuscript is more robust than the previous submission, the novel analyses seem to me to make the manuscript much more complete. I also think being more upfront and direct, as in this submission, about the role of confounding factors will be of service to other researchers interested in the field.

A couple very minor points:

p. 3 "Our group has previously reported statistically significant accuracy of classification (p=0.017)" Wouldn't the percentage accuracy be a more interesting number for the reader than the p-value? After all, the p-value is redundant with the text ("significant accuracy"), so perhaps it would be more interesting to tell the reader the classification accuracy achieved by previous studies.

p.10 "The set of regions ... in Table 2." : the sentence appears twice.

p. 13 "anecdotal evidence" The use of the term anecdotal here strikes me as incorrect. 'Anecdotal' means something that is not based on science or fact but rather on casual observation; being this conclusion the result of an experiment, it doesn't seem at all 'anecdotal.'